# Flight Phenology of *Elasmopalpus lignosellus* (Lepidoptera: Pyralidae) in the Northwest Florida Panhandle

**DOI:** 10.3390/insects14040354

**Published:** 2023-04-02

**Authors:** Abraão A. Santos, Izailda B. dos Santos, Silvana V. Paula-Moraes

**Affiliations:** West Florida Research and Education Center, Entomology and Nematology Department, University of Florida, Jay, FL 32565, USA

**Keywords:** pest fluctuation, lesser cornstalk borer, population size, pest monitoring

## Abstract

**Simple Summary:**

In this study, we documented the occurrence and abundance of lesser cornstalk borer (LCSB) male moths using a pheromone trap in the Northwest region of the Florida Panhandle. Then, we examined the relationships between the LCSB abundance and the air temperature, relative humidity, and rainfall. Our study indicated that the LCSB abundance occurs from April to December in the Northwest region of the Florida Panhandle, and the flight peak is in August. Additionally, we found that the moth abundance is positively related to increases in temperature.

**Abstract:**

*Elasmopalpus lignosellus* Zeller (Lepidoptera: Pyralidae), the lesser cornstalk borer (LCSB), is an economically important peanut pest in the southeastern U.S. region, and its occurrence and abundance have been associated with warm and dry conditions. In the Northwestern Florida Panhandle (USA), the LCSB occurrence and abundance are unknown. Thus, a study in this region used commercial sex pheromones to capture male moths year-round from July/2017 to June/2021. Our results indicated that the LCSBs were present in the region from April to December, with higher abundance in August. Moths were also caught from January to March in only 2020. In addition, the number of moths collected increased when the temperature increased. Our results indicate a different pattern for LCSB abundance than previously documented, with peak occurrence in warm and wet conditions (August). These results support that region-specific weather information should be considered when designing IPM recommendations based on the phenology of pest occurrence in the agroecosystem.

## 1. Introduction

*Elasmopalpus lignosellus* Zeller (Lepidoptera: Pyralidae), the lesser cornstalk borer (LCSB), is an economic pest of peanuts in the southeastern USA region [1,2,3]. Moths lay eggs singly or in groups (up to 120 eggs per female) on the plant or the soil surface close to the plant [4]. Once the larvae emerge, they crawl across the soil surface or on the plant, feeding on the stems of the host plants. The larval stage comprises six instars, and pupation occurs in the soil [1].

High LCSB infestation levels are associated with peanuts planted in sandy soils in hot and dry climates [1,5,6]. Economic damage can occur during the seedling and pegging stages [7]. Insecticides are the primary management option for this pest, and increased insecticide efficacy can be accomplished by refining the information about the LCSB seasonal occurrence and abundance. Pheromone traps baited with commercial sex pheromone formulations have been a valuable tool for monitoring LCSB populations, indicating periods of species occurrence and abundance associated with weather conditions [3,8].

A mathematical model indicated that weather impacts LCSB populations differently, with temperature being one of the factors that cause significant changes in the LCSB population size [5]. This species is sensitive to temperature variations, as its development is prolonged in cooler temperatures (98 days at 21 °C compared to 38 days at 27 °C) [9]. Conversely, dry climatic conditions contribute to species outbreaks [5] because rainfall can cause larval mortality, reducing the moth population’s size [1].

LCSB occurrence and abundance vary within the USA due to contrasting weather conditions in different locations. In west-central Georgia, moth activity occurs from 18 to 30 °C with three generations per year [2]. In southern Georgia, moth activity increases from early June to November [6]. In northeast Florida, moth flight was documented from March to October, when the weather conditions are usually stable over the growing season, and the temperature variations are generally less than 10 °C [3]. In South Florida, LCSB abundance occurs during the driest month (April), indicating that the species may be favored by hot and dry conditions [10]. Therefore, region-specific information on LCSB occurrence patterns and their association with weather can provide more precise timing for management decisions for this species.

Previous studies indicated dry and hot conditions are critical to LCSB abundance and outbreaks [5,10]. However, since variations in the occurrence and abundance of LCSB are mainly associated with weather conditions, it is noteworthy to determine this species’ flight phenology to provide region-specific information to support management decisions. Thus, the objectives of this study were to document, for the first time, the occurrence and abundance of LCSB moths in the Northwest region of the Florida Panhandle (Jay, FL, USA) and examine the relationships between the LCSB abundance and the air temperature, relative humidity, and rainfall.

## 2. Materials and Methods

### 2.1. Data Collection

LCSB moths were trapped year-round using a sex pheromone lure Trécé delta trap (height = 130 cm; Trécé Inc Pherocon VI trap, Adair, OK, USA) in a dryland area at the Jay Research Facility, West Florida Research and Education Center, Jay, Northwest region of the Florida Panhandle, FL (30.77622, −87.148833). The trap was examined approximately every two weeks, and the sex pheromone lure (Alpha Scents, Inc.; West Linn, OR, USA) was replaced monthly, and the pheromone trapping period ranged from July/2017 to June/2021. Our final dataset consisted of 85 trapping dates with the following number of examinations per month: 7 (January), 7 (February), 6 (March), 4 (April), 5 (May), 8 (June), 7 (July), 8 (August), 9 (September), 7 (October), 9 (November), and 8 (December).

The landscape (ca. 500 m radius) where the trapping was conducted was composed of corn, cotton, and peanut during the crop season (from March to early December), experimental plots of warm season turfgrass, and longleaf pine forest. *Brassica carinata*, wild radish, and other weed species existed during fallow. Weather data (air temperature, relative humidity, and rainfall) were acquired daily at a station (Florida Automated Weather Network, Jay Station) 824.72 m from the trap.

### 2.2. Statistical Analyses

#### 2.2.1. Circular Analysis

We performed a circular analysis to describe the flight phenology of LCSB. Three years (2018, 2019, and 2020) that presented a complete year of data collection (i.e., from January to December) were pooled and used in this analysis.

First, we transformed the dates of each collection event into their corresponding angles that represented the number of days in a year, varying from 0° (1 January) to 360° (31 December). Then, the mean angle (°) and length (0 to 1) were calculated. These metrics corresponded to the concentration of data, and length values close to 1 indicated that the data were concentrated at the same mean angle [11].

We performed the Rayleigh test to evaluate whether the circular data presented a unimodal distribution (i.e., the abundance was seasonal). Before that, we verified whether the data followed a von Mises distribution (Test = 1.23, *p* = 0.10), a premise for this test [11]. All analyses were performed using the package circular [12] in R (version 4.0.4) and R studio (version 1.2.1335) [13].

#### 2.2.2. Weather Association

Our model tested the association between weather variables (air temperature, relative humidity, and rainfall) and the moth catch. A generalized additive mixed model (GAMM) was used with the package mgcv [14], setting weather as the explanatory variable and the number of males captured on each date as a dependent variable (*n* = 85). In addition, since the samples were taken in a spaced time, a correlation structure [correlation = corAR1 (form = ~time)] was inserted to account for the potential temporal autocorrelation in the data, where time is the variable ordering the evenly spaced observations (from 1 to 85).

The average daily temperature, relative humidity, and accumulated rainfall from the previous 45 days were used to consider the period necessary for the LCSB’s development from egg to adult emergence under greenhouse conditions [6]. Initially, the Poisson distribution (link = log) was used. However, the model presented overdispersion; then, it was corrected to the quasi-Poisson family. All analyses were performed in R (version 4.0.4) and R studio (version 1.2.1335) [13]. The model suitability (concurvity, convergence, and residual distribution) was verified using the mgcv package [14]. Finally, all figures were designed using the ggplot2 package [15].

## 3. Results

### 3.1. Weather Conditions and Moth Catch

The air temperature varied from −5 °C to 30 °C during the whole period, with values higher than 18 °C from May to October (Appendix A). Changes in relative humidity were slightly higher, with average values above 70%. The lowest values registered close to 40% (Appendix A). Rainfall occurred year-round, with the highest monthly value reported in May and the lowest in December and March (Appendix A).

LCSB moths were found during all sampling periods, but the number of moths caught seemed to vary among months (Figure 1). The highest occurrence was noted in the wet and warm months (June to September), while the occurrence was not observed in cold conditions (January to March).

### 3.2. Circular Analysis

Our analysis indicated that the abundance of LCSB was seasonal (Rayleigh test = 0.69; *p* < 0.0001; Figure 2). Based on the mean vector (219.68°), August was the month with the highest abundance, with the length of the mean vector equal to 0.69 (Figure 2).

### 3.3. Weather Association

Our model had a good adjustment (R^2^ adj. = 0.66), and only temperature affected the LCSB moths’ capture (Table 1). Temperature positively contributed to the increase in moths caught (Figure 3).

## 4. Discussion

Our results indicated that the LCSB moths occurred in the Northwest region of the Florida Panhandle from April to December, with significant monthly fluctuations in abundance during the four years of pheromone trapping. During the cold months (from January to March), a reduction in the moth catch was documented, while an increase in the moth catch was observed in warm and wet conditions (from June to September). Previous research indicated that warm and dry conditions contributed to the high abundance of LCSB [3,5,10]. Here, our results showed a different pattern for LCSB moth abundance, with peaks in August, a warm and wet month in Northwest Florida.

Abiotic factors appear to be the primary drivers in regulating the population size in LCSB [1]. Mortality caused by natural enemies has less impact on the population size, and the initial egg density is the determinant factor for larval and pupal population densities [1]. The temperature has been suggested as the main driver regulating LCSB population size [5]. The lower, upper, and optimal threshold for LCSB development is estimated at 9.3 °C, 37.9 °C, and 31.39 °C, respectively [16]. This thermal range indicates that this species can adapt to cold and warm conditions with variation in the development time. Under cold conditions, the development time is prolonged (128 days at 13 °C), while in warm conditions, it is reduced (37 days at 36 °C) [16]. Moreover, diurnal cold temperatures reduce the moth activity under conditions <18 °C [2], including the oviposition rate [17]. Population outbreaks are associated with hot conditions, as LCSB is sensitive to variations in temperature, primarily cold conditions [5,17].

Other abiotic factors appear to regulate the LCSB population size but do not drive species occurrence [1]. For instance, the photoperiod does not appear to influence development [16], while rainfall tends to cause 1–2 instar mortality primarily and is a critical factor for larval population size [1]. We suggest that temperature is the primary driver of LCSB moth occurrence and abundance, supporting the previous hypothesis of temperature’s central role in population fluctuations [5]. Thus, region-specific information regarding local weather should be considered as playing a significant role in localized LCSB occurrence and abundance.

Temporal variations in weather are associated with a variation in the LCSB occurrence and abundance in the Florida Panhandle. A two-year survey in Gadsden and Jackson counties in Northeastern Florida (located 215 and 150 km from our research site, respectively) from March to October reported the occurrence of LCSB only between June and August [3]. Noteworthy, this study was the first dataset documenting the occurrence and abundance of LCSB moths in the Northwest region of the Florida Panhandle. Our temporal dataset consistently resulted in a hypothesis that variations in the local weather drive LCSB occurrence and abundance. However, at what scale the LCSB abundance can be spatially related to weather remains to be determined and deserves further investigation.

This study indicated that LCSB abundance occurred from April to December in the Northwest region of the Florida Panhandle, with peak flights in August. The LCSB is a challenging pest to be managed. The early detection and timing management of the moths is key to decreasing the damage of this pest because the larva has a borer behavior and is difficult to reach with chemical control once it enters the stem. Therefore, monitoring programs for this pest in the Northwest Florida Panhandle should be focused on periods of high temperature and humidity, especially around the middle of the season (July–August).

## Figures and Tables

**Figure 1 insects-14-00354-f001:**
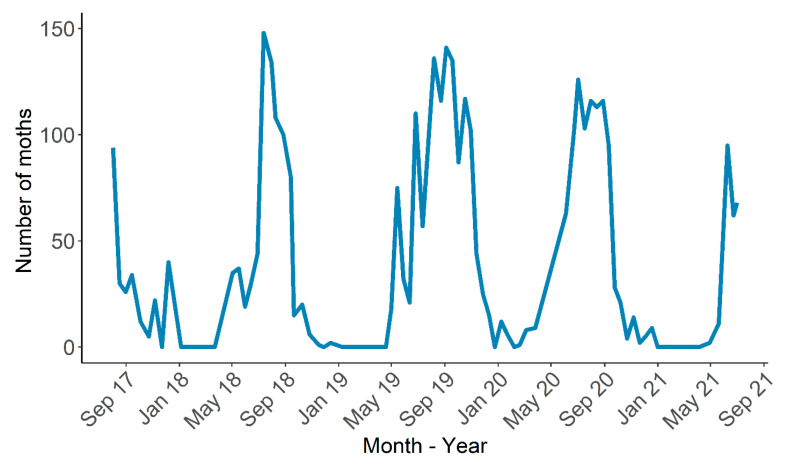
Number of LCSB male moths collected in delta traps according to the month of the year at Jay, Santa Rosa County, Florida.

**Figure 2 insects-14-00354-f002:**
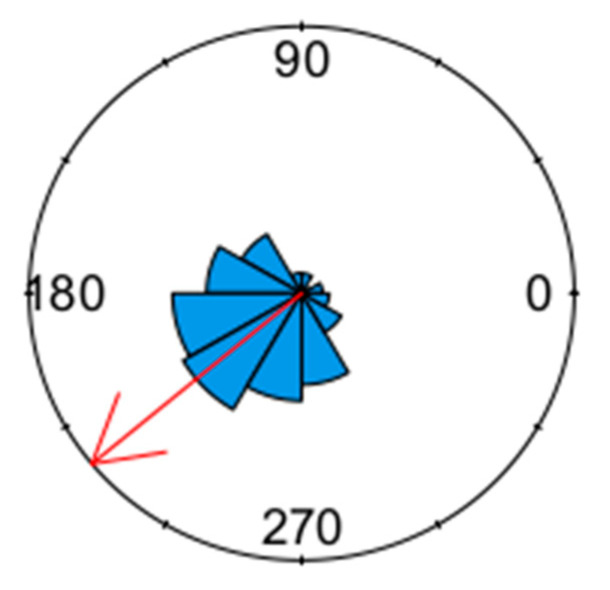
Circular plot of the number of LCSB male moths collected in delta traps at Jay, Santa Rosa County, Florida. The Rose diagram indicates the relative frequency of the moths collected across the days of the year from 0° (January 1) to 360° (December 31). The red arrow indicates the mean direction of abundance (219.68°, month = August).

**Figure 3 insects-14-00354-f003:**
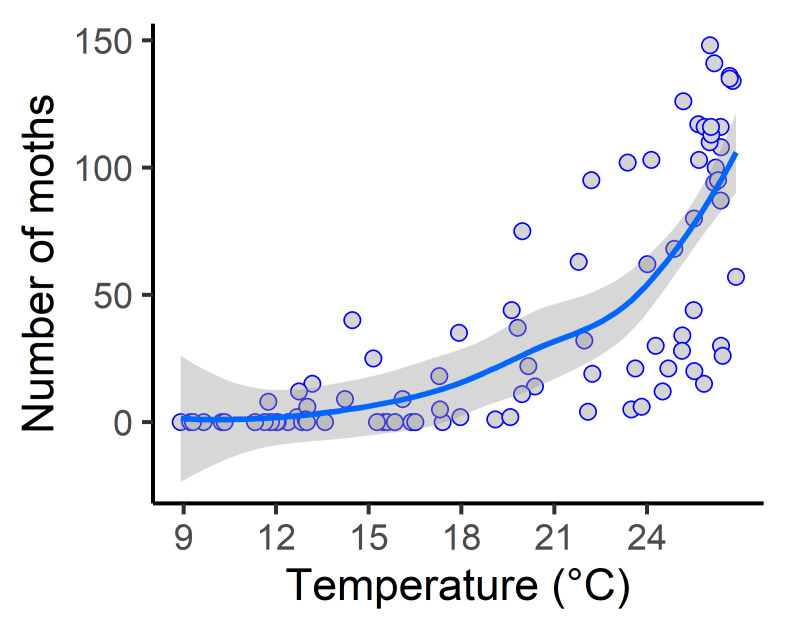
Expected counts of the LCSB male moths as a function of temperature from the previous 45 days (average) in Jay, Santa Rosa County, Florida. The blue line indicates the fitted model (GAMM with quasi-Poisson distribution) and the confidence intervals (grey color). Each blue circle represents the number of insects collected every 15 days in a delta trap for four years. (*n* = 85).

**Table 1 insects-14-00354-t001:** Summary of the generalized additive mixed models (family quasi-Poisson, link = log) for the number of LCSB moths collected and the variables of air temperature, relative humidity, and rainfall from the previous 45 days (average) before the data collection in Jay, Santa Rosa County, Florida. edf = effective degrees of freedom (*n* = 85).

**Parametric Coefficients**	**Estimated (SD)**	**t-Value**	** *p* **
Intercept	2.99 (0.19)	15.47	<0.0001
**Terms**	**edf**	**F**	** *p* **
s(Temperature)	1.00	48.09	<0.0001
s(Relative humidity)	1.00	1.97	0.16
s(Rainfall)	1.00	2.07	0.15
s(time)	0.91	0.20	0.16
R^2^ adjusted			0.66

## Data Availability

The datasets used and analyzed during the current study are available from the corresponding author upon reasonable request.

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
