# Peer review of "Flight Phenology of Elasmopalpus lignosellus (Lepidoptera: Pyralidae) in the Northwest Florida Panhandle"

_insects, 2023, doi:10.3390/insects14040354_

Round 1
Reviewer 1 Report
The authors studied the population dynamics of male Elasmopalpus lignosellus over three years using pheromone traps and found that temperature and relative humidity affected their dynamics in the Florida Panhandle US.
An important shortcoming in this paper is that the objectives are unclear. Is the LCB ‘only a material’ for the purpose of studying the effects of climate change on the population dynamics of insects in the climate transition zone, or do you want to study the ecology of the LCB in the Florida Panhandle?
Here are the specific revisions
Simple Summary
L9. The connection between the first line and the second line is unclear and should be removed. I don’t know what size it is.
L13. Regarding the abbreviations, I checked the papers and found 11 for LCSB, while 186 for LCB, the latter being the major one, so LCB should be used.
L14. As mentioned before, the objective is unclear.
L18-19. I don’t know which point is ‘new insight’.
Abstract
L23-25…abundance are unknown, thus a study was conducted in the northwestern Florida Panhandle (Jay, FL) using commercial sex pheromones to capture male moths year-round.
L25-26. Delete.
Introduction
It should be clearly stated that only males enter the trap and that the population dynamics of males are being studied.
What is ‘Flight phenology’?
Are there any other similar studies using other insects in climate transition zone.
M & M
L100-101. Why is the number of traps different among months?
Results
L154. Figure 1. These figures should be move to appendix or remove. Data were not taken by the authors, and it is hard to understand when looking at the raw data.
L165-166. Convert U and P to italics
L164-165. Were you tested independently between years rather than post-hoc tested? If so, please use correct statistical methods.
L188. What about interactions?
Discussion
L200-203. Please discuss why this is different from previous research.
L208-213. These sentences should be move to introduction.
L219-222. Does that mean that after all, with previous studies, it doesn’t matter if th e climate transition zone is or not?
L228-229. While this may be the first data for the Florida population, there are already studies using the same material in other populations. In this respect, it lacks originality.
L231-232. I don’t understand the mean of this sentence.
L235. There is little discussion of relative humidity.
Please discuss specifically how this can be used for pest control and forecasting.
Author Response
#Reviewer1
The authors studied the population dynamics of male Elasmopalpus lignosellus over three years using pheromone traps and found that temperature and relative humidity affected their dynamics in the Florida Panhandle US.
An important shortcoming in this paper is that the objectives are unclear. Is the LCB ‘only a material’ for the purpose of studying the effects of climate change on the population dynamics of insects in the climate transition zone, or do you want to study the ecology of the LCB in the Florida Panhandle?
>>We thank you, the reviewer, for pointing out the unclear message in our manuscript. Our objective was to document the moth occurrence and abundance in the Florida Panhandle, as this area is of importance for pest populations in the U.S. Please, notice that we have changed the title to match our objective and made other corrections throughout the manuscript as suggested by the reviewer.
Here are the specific revisions
Simple Summary
L9. The connection between the first line and the second line is unclear and should be removed. I don’t know what size it is.
>>We have altered this sentence, please lines 9-10.
L13. Regarding the abbreviations, I checked the papers and found 11 for LCSB, while 186 for LCB, the latter being the major one, so LCB should be used.
>>This variation is due to the recent suggestion by the Entomological Society of America to use LCSB instead of LCB since the first one included the acronym for stalk (S). You can check the common name list at https://www.entsoc.org/publications/common-names
L14. As mentioned before, the objective is unclear.
>> The section was reorganized to be clearer and objective is mentioned in lines 9-10.
L18-19. I don’t know which point is ‘new insight’.
>>This information was removed. Please see lines 13-14.
Abstract
L23-25…abundance are unknown, thus a study was conducted in the northwestern Florida Panhandle (Jay, FL) using commercial sex pheromones to capture male moths year-round.
>>We have modified the sentence to include the reviewer’s suggestions. Please see lines 17-19
L25-26. Delete.
>>The sentence was deleted. Please see line 20.
Introduction
It should be clearly stated that only males enter the trap and that the population dynamics of males are being studied.
>>By pheromone trap, it means that only male is attracted (trapped) since only moth females’ pheromone is currently produced for commercial use. Therefore, this mention is irrelevant because there is no “male pheromone” in the studied system.
What is ‘Flight phenology’?
>>Flight phenology refers to the dynamic of flight occurrence/abundance though time. This is a standard terminology used in Entomology to describe this type of study. Please see the references:
Kaspari, M., Pickering, J. and Windsor, D. (2001), The reproductive flight phenology of a neotropical ant assemblage. Ecological Entomology, 26: 245-257. https://doi.org/10.1046/j.1365-2311.2001.00320.x
Valtonen, A., Ayres, M.P., Roininen, H. et al. Environmental controls on the phenology of moths: predicting plasticity and constraint under climate change. Oecologia 165, 237–248 (2011). https://doi.org/10.1007/s00442-010-1789-8
Merckx, T., Nielsen, M.E.; Heliola, J., Kuusaari, M., Pettersson, L.B.; Poyry, J., Gotthard, K., Kivela, S.M. (2021). Urbanization extends flight phenology and leads to local adaptation of seasonal plasticity in Lepidoptera. Proceedings of the National Academy of Sciences 118 (40) e2106006118. https://doi.org/10.1073/pnas.2106006118
Muñoz-Adalia, E. J., Ahmed, J., & Colinas, C. (2022). Microclimatic conditions drive summer flight phenology of Platypus cylindrus in managed cork oak stands. Journal of Applied Entomology, 146, 964– 974. https://doi.org/10.1111/jen.13025
Are there any other similar studies using other insects in climate transition zone.
>Regarding agricultural pests in our area, these studies have been conducted for the first time and are ahead for future publications. We have also removed the mention of the climate transition zone and kept the manuscript focused on forecasting LCSB and its implications for pest management programs.
M & M
L100-101. Why is the number of traps different among months?
>> This happened due to logistic reasons, in which it was not possible to access the trap (weather conditions, such as hurricane season). Plus, this refers to the number of times that the trap was examined per month. Please see lines 76-79.
Results
L154. Figure 1. These figures should be move to appendix or remove. Data were not taken by the authors, and it is hard to understand when looking at the raw data.
>>This dataset is important because informs the weather conditions during the study period. However, we agree that it could be moved to the appendix and did so. Please see the appendix.
L165-166. Convert U and P to italics
>>We kept the current form because there is no guideline regarding the writing style of P-values in the MPDI instruction on authors’ webpage.
L164-165. Were you tested independently between years rather than post-hoc tested? If so, please use correct statistical methods.
>>We thank the reviewer for indicating such shortcoming in our data that was not previously considered. Because there is no test to verify if the two samples are independent or not in circular analysis, and the assumption of independence disagrees with the temporal analysis performed for association with environmental variables, we removed this test and pooled the data from the three years to calculate an overall trend of LCSB in the area studied. Please see lines 90-102.
L188. What about interactions?
>>We tried to; however, the model didn’t converge. Thus, we followed the parsimony principle to keep the simple model with the highest explanation.
Discussion
L200-203. Please discuss why this is different from previous research.
>>We have indicated that “Previous research indicated that warm and dry conditions contribute to high abundance of LCSB [3,5,10]. Here, our results showed a different pattern for LCSB moth abundance, with peaks in August, a warm and wet month in Northwest Florida.” Please see lines 203-206.
L208-213. These sentences should be move to introduction.
>These sentences are part of the main argument in the paragraph regarding the effect of temperature on population.
L219-222. Does that mean that after all, with previous studies, it doesn’t matter if the climate transition zone is or not?
>>As we mentioned previously, we rewrite the manuscript to focus on the implications of forecasting LCSB for management plans.
L228-229. While this may be the first data for the Florida population, there are already studies using the same material in other populations. In this respect, it lacks originality.
>> Our arguments claim that “region-specific weather information should be considered when documenting the phenology of pests in the agroecosystem to support precise timing of management decisions in IPM programs for LCSB.” In that sense, our manuscript presents this perspective for the northeastern Florida Panhandle, which is done for the first time. This region-specific study type is relevant and other examples been published in the Insects journal. For instance:
Swengel, A.B.; Swengel, S.R. Complex Messages in Long-Term Monitoring of Regal Fritillary (Speyeria idalia) (Lepidoptera: Nymphalidae) in the State of Wisconsin, USA, 1988–2015. Insects 2017, 8, 6. https://doi.org/10.3390/insects8010006
Lin, C.-Y.; Batuman, O.; Levy, A. Identifying the Gut Virome of Diaphorina citri from Florida Groves. Insects 2023, 14, 166. https://doi.org/10.3390/insects14020166
Ceia-Hasse, A.; Boieiro, M.; Soares, A.; Antunes, S.; Figueiredo, H.; Rego, C.; Borges, P.A.V.; Conde, J.; Serrano, A.R.M. Drivers of Insect Community Change along the Margins of Mountain Streams in Serra da Estrela Natural Park (Portugal). Insects 2023, 14, 243. https://doi.org/10.3390/insects14030243
Amarasekare, K.G.; Link, R.H. Abundance of Halyomorpha halys (Hemiptera: Pentatomidae) and Megacopta cribraria (Hemiptera: Plataspidae) in Soybean in Areas with Few Previous Sightings in Tennessee. Insects 2023, 14, 237. https://doi.org/10.3390/insects14030237
Joshi, N.K.; Phan, N.T.; Biddinger, D.J. Management of Panonychus ulmi with Various Miticides and Insecticides and Their Toxicity to Predatory Mites Conserved for Biological Mite Control in Eastern U.S. Apple Orchards. Insects 2023, 14, 228. https://doi.org/10.3390/insects14030228
Zhao, L.; Gao, R.; Liu, J.; Liu, L.; Li, R.; Men, L.; Zhang, Z. Effects of Environmental Factors on the Spatial Distribution Pattern and Diversity of Insect Communities along Altitude Gradients in Guandi Mountain, China. Insects 2023, 14, 224. https://doi.org/10.3390/insects14030224
L231-232. I don’t understand the mean of this sentence.
>>We rewrite this sentence to make it clear. Please see lines 232-235.
L235. There is little discussion of relative humidity.
>>We believe that discussing relative humidity is irrelevant as the only temperature seems associated with moths catch in our model.
Please discuss specifically how this can be used for pest control and forecasting.
>> We have added in lines 236-242: Overall, this study indicated that LCSB occurs from April to December in the North-west region of the Florida Panhandle, with peak flights in August. LCSB is a challenging pest to be managed. Early detection and timing management of moths are key aspects to decrease the damage of this pest, because larva has a borer behavior and is difficult to be reached in chemical control once it enters the stem. Monitoring programs for this pest in the Northwest Florida Panhandle should be focused on periods of high temperature and humidity, especially around the middle of the season (July-August).
Reviewer 2 Report
This paper discusses the effects of temperature and humidity on the occurrence of the butterfly Elasmopalpus lignosellus (an agricultural peanut pest) in Florida Panhadle.
Overall, this paper could be improved and attract more readers if the authors could emphasize more the broader relevance, beyond agriculture. To this end, the introduction and the discussion need to be adapted and more literature would have to mentioned. In addition, I advice to discuss the underlying mechanism for the preferences of the butterfly in a more detailed manner, taking environmental and physiological aspects into account. As it stands the paper is mainly descriptive instead of explanatory.
Details:
Line 27: "... similar number...". This is not very informative if the number is not quantified.
Line 29: It may not be clear to many readers what "weather number" means.
Section 2.2.1. To avoid any misunderstanding, I would call this section "Statistical analysis of the circular data set". Overall, the explanation of this section could be improved. 'X' seems to be a "time interval" with dimension Time (not a "time unit"). The unit of the time interval is e.g. hour, day, week, or month. A similar reasoning holds for k which also has the dimension of time. It seems to correspond to a full year. The 360 could be replaced by the constant alpha (=360 degrees) to allow for the correct units. It should also be made explicit whether parameter a is continuous, or whether it can only adopt discrete values. Why is it necessary to put the 360 and X between brackets? I recommend to use italics for the symbols in the formulae. If one computes a vector it means that both the amplitude (length) and direction are essentially known. Perhaps, the authors mean the parameter (i) is the direction of the mean vector rather than the mean vector (line 118). A similar reasoning holds for the median vector. Parameter (iv) indicates (rather than measuring it) the circular variability.
The model does not explain anything; It shows correlations with confidence intervals. The explanation of the observed correlations (e.g. based on the physiological profile of the butterflies) is a weak aspect of the paper.
Author Response
#Reviewer 2
This paper discusses the effects of temperature and humidity on the occurrence of the butterfly Elasmopalpus lignosellus (an agricultural peanut pest) in Florida Panhadle. Overall, this paper could be improved and attract more readers if the authors could emphasize more the broader relevance, beyond agriculture. To this end, the introduction and the discussion need to be adapted and more literature would have to mentioned. In addition, I advice to discuss the underlying mechanism for the preferences of the butterfly in a more detailed manner, taking environmental and physiological aspects into account. As it stands the paper is mainly descriptive instead of explanatory.
>>We welcome and acknowledge the reviewer’s comments and suggestions. We have directed the main message of the manuscript to the context of pest management of the agriculture pest, as it fits the section for which this communication was sent.
We disagree with the reviewer regarding the lack of environmental and physiological aspects discussion. Please, see lines 43-57 (introduction) and 207-226 (discussion), which we explain the current association of this moth species (not a butterfly, as mentioned) with the environment, physiological limits for species development, and how together their impact this species occurrence and abundance across the southeastern US.
Details:
Line 27: "... similar number...". This is not very informative if the number is not quantified.
>>The sentence was corrected to be clear. Please see lines 20-21.
Line 29: It may not be clear to many readers what "weather number" means.
>>The sentence was rewritten to be concise. Please see lines 20-21.
Section 2.2.1. To avoid any misunderstanding, I would call this section "Statistical analysis of the circular data set". Overall, the explanation of this section could be improved. 'X' seems to be a "time interval" with dimension Time (not a "time unit"). The unit of the time interval is e.g. hour, day, week, or month. A similar reasoning holds for k which also has the dimension of time. It seems to correspond to a full year. The 360 could be replaced by the constant alpha (=360 degrees) to allow for the correct units. It should also be made explicit whether parameter a is continuous, or whether it can only adopt discrete values. Why is it necessary to put the 360 and X between brackets? I recommend to use italics for the symbols in the formulae. If one computes a vector it means that both the amplitude (length) and direction are essentially known. Perhaps, the authors mean the parameter (i) is the direction of the mean vector rather than the mean vector (line 118). A similar reasoning holds for the median vector. Parameter (iv) indicates (rather than measuring it) the circular variability.
>> We have changed the circular analysis based on reviewer #1 comments and explain how it was performed:
“We performed a circular analysis to describe the flight phenology of LCSB. Three years (2018, 2019, and 2020) that presented a complete year of data collection (i.e., from January to December) were pooled and used in this analysis.
First, we transformed the dates of each collection event in their corresponding angles that represent the number of days in a year, varying from 0° (January 1 to 360° (December 31). Then, the mean angle (°) and length (0 to 1) were calculated. These parameters correspond to the concentration of data, and length values close to 1 indicate that the data is concentrated at the same mean angle.
We performed the Rayleigh test to evaluate if the circular data presents a unimodal distribution (i.e., the abundance is seasonal). Before that, we verify if the data follows VonMisses distribution (Test = 1.23, p = 0.10), a premise for this test [21]. All analyses were performed using the package circular [19] in R (version 4.0.4) and R studio (version 1.2.1335) [20].”
We believe the additional mention of “Statistical analysis of the circular data set” is unnecessary since section 2.2. indicate Statistical analyses and 2.2.1 Circular analysis.
The model does not explain anything; It shows correlations with confidence intervals. The explanation of the observed correlations (e.g. based on the physiological profile of the butterflies) is a weak aspect of the paper.
>>As we mentioned, we disagree with such a mention. In lines 43-57 (introduction) and 207-226 (discussion), we explain the current association of this moth species with the environment, physiological limits for species development, and how together their impact this species occurrence and abundance across the southeastern US.
Although models represent an estimation of an event, with a level of uncertainty associated, they are useful tools for IPM. Our communication indicates that increases in temperature contribute to this moth occurrence and abundance, which would potentially benefit pest management plans. This association is a central core for field studies regarding weather and insect population size. For instance, please see:
Kaspari, M., Pickering, J. and Windsor, D. (2001), The reproductive flight phenology of a neotropical ant assemblage. Ecological Entomology, 26: 245-257. https://doi.org/10.1046/j.1365-2311.2001.00320.x
Valtonen, A., Ayres, M.P., Roininen, H. et al. Environmental controls on the phenology of moths: predicting plasticity and constraint under climate change. Oecologia 165, 237–248 (2011). https://doi.org/10.1007/s00442-010-1789-8
Merckx, T., Nielsen, M.E.; Heliola, J., Kuusaari, M., Pettersson, L.B.; Poyry, J., Gotthard, K., Kivela, S.M. (2021). Urbanization extends flight phenology and leads to local adaptation of seasonal plasticity in Lepidoptera. Proceedings of the National Academy of Sciences 118 (40) e2106006118. https://doi.org/10.1073/pnas.2106006118
Muñoz-Adalia, E. J., Ahmed, J., & Colinas, C. (2022). Microclimatic conditions drive summer flight phenology of Platypus cylindrus in managed cork oak stands. Journal of Applied Entomology, 146, 964– 974. https://doi.org/10.1111/jen.13025